# Dynamical Behavior of the Human Ferroportin Homologue from *Bdellovibrio bacteriovorus*: Insight into the Ligand Recognition Mechanism

**DOI:** 10.3390/ijms21186785

**Published:** 2020-09-16

**Authors:** Valentina Tortosa, Maria Carmela Bonaccorsi di Patti, Federico Iacovelli, Andrea Pasquadibisceglie, Mattia Falconi, Giovanni Musci, Fabio Polticelli

**Affiliations:** 1Department of Sciences, Roma Tre University, 00146 Rome, Italy; valentina.tortosa@uniroma3.it (V.T.); andrea.pasquadibisceglie@uniroma3.it (A.P.); 2Department of Biochemical Sciences, Sapienza University of Roma, 00185 Rome, Italy; mariacarmela.bonaccorsi@uniroma1.it; 3Department of Biology, University of Rome Tor Vergata, 00133 Rome, Italy; federico.iacovelli@uniroma2.it (F.I.); falconi@uniroma2.it (M.F.); 4Department Biosciences and Territory, University of Molise, 86090 Pesche, Italy; giovanni.musci@unimol.it; 5National Institute of Nuclear Physics, Roma Tre Section, 00146 Rome, Italy

**Keywords:** ferroportin, *Bdellovibrio bacteriovorus*, major facilitator superfamily, molecular dynamics, iron transporter

## Abstract

Members of the major facilitator superfamily of transporters (MFS) play an essential role in many physiological processes such as development, neurotransmission, and signaling. Aberrant functions of MFS proteins are associated with several diseases, including cancer, schizophrenia, epilepsy, amyotrophic lateral sclerosis and Alzheimer’s disease. MFS transporters are also involved in multidrug resistance in bacteria and fungi. The structures of most MFS members, especially those of members with significant physiological relevance, are yet to be solved. The lack of structural and functional information impedes our detailed understanding, and thus the pharmacological targeting, of these transporters. To improve our knowledge on the mechanistic principles governing the function of MSF members, molecular dynamics (MD) simulations were performed on the inward-facing and outward-facing crystal structures of the human ferroportin homologue from the Gram-negative bacterium *Bdellovibrio bacteriovorus* (BdFpn). Several simulations with an excess of iron ions were also performed to explore the relationship between the protein’s dynamics and the ligand recognition mechanism. The results reinforce the existence of the alternating-access mechanism already described for other MFS members. In addition, the reorganization of salt bridges, some of which are conserved in several MFS members, appears to be a key molecular event facilitating the conformational change of the transporter.

## 1. Introduction

The major facilitator superfamily (MFS) represents the largest class of secondary active transporters [1,2]; its members are found in bacteria, archaea, and eukarya [3]. MSF members transport a wide range of substrates, including inorganic anions and cations, amino acids, peptides, monosaccharides, oligosaccharides, enzyme cofactors, toxins, drugs and iron chelates [4]. They operate by uniport, symport, or antiport mechanisms [2]. The transporters share a conserved and characteristic fold, called the “MFS fold”, a canonical 12 transmembrane segments (12-TM) fold consisting of two domains, known as the N- and the C-domains. Each domain is organized into a pair of inverted “3 + 3” repeats, connected by a cytoplasmic loop [2]. In the N-domain, TMs 1, 2 and 3 are related to TMs 4, 5 and 6 by an approximate 180° rotation around an axis parallel to the membrane bilayer. A similar relationship takes place in the C-domain between TMs 7, 8, 9 and TMs 10, 11 and 12 [5]. Two conserved sequences are found at the cytoplasmic ends of TMs 2 and 3 (N-domain) and of TMs 8 and 9 (C-domain) in many MFS members [6]. These short connecting segments join the ends of the TM (TM2-TM3; TM8-TM9) thus allowing better control of relative motions on the cytoplasmic side [6]. Conversely, the loop linking the N- and C-domains is usually long in order to support a high degree of relative motion between the two domains during the transport cycle [7]. Despite the difference in size, shape and chemical properties of their substrates, and in the transport mode of the proteins (symport, antiport, and uniport), MFS members often contain several conserved motifs that play important roles in common functions of MFS. In particular four motifs, A, B, C and D, have been identified in MFS transporters [8,9].

Motif-A [G-(X3)-(D/E)-(R/K)-X-G-[X]-(R/K)-(R/K)] (where ‘X’ indicates any amino acid) is the most conserved signature in MFS transporters; it is located between the TM2 and TM3 in the N-domain (loop 2-3, L2-3). The mail role of the motif-A is the stabilization of the outward-facing state and the charge-relay triad is a key player of this regulation. The conformational change is induced and orchestrated via an inter-domain linker (L6–7), containing a number of polar residues and an amphipathic α-helix, which probably binds to the membrane surface [10].

Motif-B, “RXXQG” in TM4 was proposed to be involved in energy coupling and substrate binding-induced conformational change in MFS transporters [11], although the precise mechanisms for its action remains to be elucidated.

Motif-C in the rocker-helix TM5 [G-(X6)-G-(X3)-GP-(X2)-GP-(X2)-G] is necessary for the drug/H+ antiport activity (the antiporter motif) [12]. It has been proposed that motif-C has specifically evolved to stabilize the hydrophobic inter-domain interaction of MFS antiporters in the inward-facing state, in which an antiporter binds its substrate [9].

Motif-D is only found in some subgroups of MFS transporters, including the MdfA multidrug efflux protein and its orthologues [9]. It is not yet clear what role motif-D plays in the transport activity of these transporters [9]. However, several point mutations in motif-D have been analyzed experimentally, demonstrating that this motif is important for MdfA functions [9,13,14,15,16]. Motif-D in TM1 of MdfA contains two acidic residues essential for transport activity (E25 and D34). These two residues are the candidate protonation sites of MdfA [9,15].

Distinct conformational shifts are necessary to ensure the substrates transport across membranes [2,5]. In a seminal paper, Jardetzky in 1966 proposed for the first time the “alternate access model”. According to this model, when the substrate binds on one side of the membrane, the transporter undergoes a conformational change allowing the protein to release the substrate on the opposite side, thereby allowing it to cross the lipid bilayer. During the transport cycle, the binding site is never simultaneously exposed to both sides of the biological membrane, preventing the formation of a channel-like structure that would result in the free diffusion of the substrate (leak) [17]. Thus, to complete a transport cycle, at least three different conformational states of the protein would be required: inward-facing, occluded and outward-facing [2,5]. Structural and functional analyses of the MFS transporters largely confirmed this model, showing that the alternating access cycle is mediated by a so-called “rocker-switch” movement that operates rigid-body rotation of the two domains [2,6]. During this rigid-body motion the first helix (TM1, 4, 7, and 10) and second helix (TM2, 5, 8, and 11) in each repeat shift their interaction along the cytoplasmic ends in the outward-facing state to an interaction along the extracellular ends in the inward-facing states. Changes in salt-bridges networks could play a key role in mediating the conformational changes observed in these transporters [2,6,18]. However, the existence of occluded structures suggests that the rocker-switch model, involving only a rigid-body rotation of the two domains, may not be sufficient to describe the conformational changes occurring in MFS members during the transport cycle. Therefore, in 2016 Quistgaard and colleagues proposed a revision of the rocker-switch model [2]. The authors described a “clamp-and-switch model” in which the transition between the different states involves not only rigid-body rotations, but also structural changes in individual transmembrane helices [2].

A member of the MFS that is relevant from a biomedical viewpoint is human ferroportin (hFpn), the only human iron exporter identified so far [19]. hFpn mutations are the cause of type 4 haemochromatosis, a disease characterized by two different iron accumulation phenotypes [20]. These different phenotypes are the result of loss-of-function mutations, which impair the transport activity, or gain-of-function mutations, which impair hepcidin-mediated hFpn internalization and degradation [21]. Very recently, a 3.2 Å cryo-EM structure of hFpn in the inward-facing conformation has been described in a manuscript posted on the popular preprint server for Biology bioRxiv (PDB code: 6W4S), although the atomic coordinates have not been made available yet [22]. Thus, no high-resolution structure for this protein is available in any conformational state.

To gain further detailed insight in the molecular features underlying the conformational changes in MFS members during the transport cycle, MD simulations have been performed on the available high resolution crystal structures of a bacterial homologue of hFpn from the predatory Gram-negative bacterium *Bdellovibrio bacteriovorus* (BdFpn) [23,24]. Indeed, this is one of the few MFS transporters with a human homologue whose crystal structure is available in both the inward-facing and outward-facing states. Further, in the absence of high-resolution crystal structures of hFpn, the study of BdFpn provides the unique chance to investigate the dynamical properties of an iron transporter of the MFS.

## 2. Results

The dynamical behavior of BdFpn embedded in a lipid bilayer was studied through molecular dynamics simulations of three different systems: BdFpn in the inward-facing and outward-facing states, to probe the global dynamic behavior of the transporter (Inward_Apo and Outward_Apo, respectively), and BdFpn in the inward-facing state with an excess of iron ions (Inward_Fe), aimed at investigating the encounter of iron with protein residues in the substrate binding pocket, this being the initial event of the ferroportin-mediated iron efflux mechanism. In addition, to test the reproducibility of the results, three independent, 200 ns long, MD simulations were performed for each system (Table 1).

### 2.1. Inward_Apo

The inward-facing structure of BdFpn in the apo form (PDB: 5AYO) [24] was subjected to MD simulations first to probe the conformational stability of the protein and to establish a baseline for its dynamical behavior. Analysis of the RMSD values, describing the evolution of the sampled conformations in terms of distance from the starting structure, indicates that the system reaches the stability after only 25 ns of simulation time. The absence of large displacements (RMSD average values around 2.5 Å) indicates that the modelled loop (see Methods for details) does not impair the behavior of the protein, and that this model is suitable for further studies (Appendix A).

Principal component analysis (PCA) [25] was performed to identify the dominant modes of motion characterizing BdFpn (Figure 1). The projection of the dominant motion on the three-dimensional structure of BdFpn indicates that the protein undergoes a non-symmetric rocker-switch movement, in agreement with the hypothesized alternating-access mechanism [5]. These data indicate that BdFpn in the inward-facing conformation is characterized by a specific displacement of the helices towards the intracellular side, as if the protein would tend to assume a more occluded conformation. This result suggests that the outward-facing conformation may be more energetically favorable in the BdFpn apo-state.

To investigate the functional involvement of specific protein residues during the transport process, the salt bridges network characterizing the inward-facing conformation of the transporter has been analyzed. As shown in Figure 2 the following interactions have been identified:D162-R17—the interaction between D162 (TM 5) and R17 (TM 1) is present in 93.6% of the simulation time for the three replicas;D229-R371—the salt-bridge between the D229 (loop TM6-TM7) and R371 (TM 11) can be observed in 97.1% of the simulation time for the three replicas;D28-R284—the salt-bridge between D28 and R284, present for about 95.8% of the simulation time for the three replicas, is located in the upper TM region and stabilizes the interaction between the N (TM1) and C (TM8) domains;E10-K150—this salt bridge, present in 52.5% of the simulation time for the three replicas, connects the C-terminal end of TM1 with the N-terminal end of TM5;E277-H180—this interaction (present in about 62.8% of the simulation time for the three replicas) appears to be essential for establishing the interaction of L-TM5-TM6 with TM8, characterizing the inward-facing conformation.

### 2.2. Outward_Apo

In order to get a deeper understanding of the global dynamic behavior and of the conformational rearrangements of the iron transporter, MD simulations of BdFpn in the outward-facing state were performed using the same protocol followed for the inward-facing conformation. The results of the simulations were also interpreted in comparison with the inward-facing structure simulations. Analysis of the RMSD values indicates that the system reaches the stability after 60–70 ns of simulation time. The absence of large displacements (RMSD values ranging from 2.5 to 3.0 Å) suggests that the structure samples a restricted conformational basin (Appendix A). 

The projection of the dominant motion on the three-dimensional structure of the outward-facing state indicates that the protein motions are reduced in this conformation in comparison with the inward facing one (Figure 3). This is a further evidence that the outward-facing conformation could actually be more energetically favorable in the BdFpn apo-state.

In addition, analysis of the essential subspace defined by the projection of the first two eigenvectors for the two systems indicates that the apo form of the inward-facing state samples a wider area with respect to the outward-facing one (Figure 4). These results confirm that the fully opened outward-facing protein structure samples a restricted conformational basin, supporting the hypothesis that this conformation is the energetically favored one.

The stability of the outward open conformation can be explained by the presence of several salt bridges and hydrogen bonds. In fact, the analysis of these features highlights how the residues that mark the intracellular end of the protein lock the transporter in this conformation. In particular, the D140 residue at the N-terminal end of TM4 closely interacts with two arginine residues, R371 (interaction present in about 98.5% of the simulation time) on the C-terminal end of TM8 and R73 on the N-terminal end of TM3 (present in about 99% of the simulation time). D140 also interacts with K219, located in the long intracellular loop between TM6 and TM7 (present in about 99% of the simulation time). Moreover, a network of transient interactions is observed between D241 (at the end of the L TM6-TM7) and R370 (at the end of L TM10-TM11) (present in about 99.8% of the simulation time), between E373 and K228 (L TM6-TM7) (present in about 100% of the simulation time), and between D69 (at the C-terminal end of TM2), and again K228 (present in about 83% of the simulation time). Other important interactions were found between E149 (N-terminal end of TM5) which forms a salt bridge with R364 (C-terminal end of TM10) (present in about 53% of the simulation time), and E203 (TM6) forming a salt bridge with R17 (TM1) (present in about 99% of the simulation time) (Figure 5).

### 2.3. Inward_Fe

To investigate the role of protein residues in iron binding and translocation, three independent MD simulations of the inward-facing BdFpn structure with an excess of Fe^2+^ ions (Inward_Fe system) were performed. The convergence of the trajectories of the protein in the presence of iron ions was checked by evaluating the results of RMSD analysis (Appendix A), which shows deviations ranging from 2.5 Å to 3.0 Å for all the simulations, indicating that the presence of the excess of iron ions does not significantly alter the protein’s structure over the simulation time. 

In two out of three simulations an influx of iron ions within the protein’s cavity is observed, with preferential location around the N-domain, which appears to serve as the primary iron-recognition site in BdFpn (Figure 6).

The cavity-facing sides of the N- and C-domains have contrasting surface electrostatic potential values with a patch of negative electrostatic potential along the central cavity on the N-domain (Figure 7). It is probably this electrostatic potential gradient that drives iron entrance and localization into the transporter’s cavity.

The analysis of iron-occupancy during the MD simulations allowed the identification of residues potentially involved in iron coordination (Table 2 and Appendix A).

The polarity of the D24, D134, S130, E203 and D229 residues provides a suitable “hot-spot” for iron recognition (yellow spheres in Figure 7). Overall, the analyses carried out suggest a multisite model for the metal translocation in which iron diffuses into the protein’s cavity through a network of key residues regulating substrate translocation.

## 3. Discussion

Different MD simulations starting from BdFpn crystal structures in the inward-facing and outward-facing conformational states were performed. The aim of these simulations was to provide dynamics information about the global behavior of BdFpn, focusing on the protein regions that could be directly involved in the coordinated motion of the “alternating access mechanism” and on the dynamical interactions between the transporter and iron. The analyses indicate that the outward-facing conformation is the more energetically favorable one in the BdFpn apo state, with a tendency of the apo inward-facing structure to close at the intracellular side. Such a behavior is also confirmed by the comparison of the analyses carried out on the trajectory of the outward-facing apo state, which revealed a reduction of its essential conformational subspace (Figure 4). 

MD simulations confirm that the reorganization of inter- and intra-domain salt bridges is important in controlling the helical motions required for the conformational change between the two conformational states [2]. In the outward-facing state the N-terminal of TM5 juxtaposes the C-terminal of TM8 and TM10, the N-terminus of TM11 moves between the C-termini of TM2 and TM4 and the N-terminal of TM9 shifts between the C-terminal of TM8 and C-terminal of TM10. These contacts are supported by interactions between the side chains of E149 (N-terminal of TM5) and R364 (C-terminal of TM10), and between E203 (TM6) and R17 (TM1). D69 (TM2), from conserved motif A, is part of an electrostatic cluster involving E373, R370, D241 and a residue of the intermediate loop between TM6 and 7 (Lys228). An adjacent cluster, including R371, D140, R73 and K219, located in the long intracellular loop between helices 6 and 7, is also observed. 

When the transporter switches in the inward-facing state, the two lobes rotate largely as rigid bodies around an axis parallel to the plane of the membrane bilayer. The interactions described above are replaced by contacts between the extracellular side halves of TMs 1, 2 and 5 of the N-terminal domain, and TMs 7, 8, and 11 of the C-terminal domain. This is prompted by a sliding of TM11 along TM2 and a significant rearrangement of the loop between TM5-TM6, leading to position TM7 near TM8 and TM1. This rearrangement results in a closure of the gate between the domains on the other side of the membrane (Appendix A; panel B). 

Similar interactions can also be observed in other MFS transporters. The “gating residues” alternately break and reform interactions during the transport cycle, as a response to the reciprocal structural changes, allowing the protein to progress along different conformational states [2]. Specific residues that mediate interactions between the N-domain and the C-domain are important during the conformational cycle, as well as for the binding of the substrate, which may function much like allosteric regulators of enzymes in modifying the energy landscapes of the different conformational states [2]. The conserved motif-A (D69-R73-D140 in BdFpn) seems to be important for stability and transport activity. The analysis of all known MFS transporters structures has shown that the motif-A forms different gating interactions in the inward-facing, occluded and the outward-facing states [2].

It must also be stressed that D69, R73, D140, D162 and R371 in BdFpn are all conserved in hFpn (D84, R88, D157, D181 and R489, respectively) and are related to hereditary iron disorders [27]. This confirms the functional importance of these residues in the Fpn transport activity. Recent work on hFpn confirms the importance of interaction networks involving salt bridges between these residues to form the intracellular gate [22,28].

Molecular mechanisms underlying iron transport across the membrane are not well understood. To investigate the role of protein residues in the iron binding and translocation, MD simulations with an excess of Fe^2+^ ions have also been performed. In two out of the three simulations, an influx of iron ions within the protein’s cavity is observed. The analyses allowed us to identify residues that could be important for iron translocation through the transporter. Iron mainly interacts with residues belonging to the N-domain in line with observations obtained for most of the MFS transporters [5]. The simulations capture spontaneous iron translocation inside the cavity with preferential location around D24, D134, S130, E203 and D229 residues, supporting the view that D24 is involved in iron transport [23,24]. It must be underlined that the residue corresponding to D24 in hFpn (i.e., D39) was previously reported to be important for both substrate binding and transport [19]. However, D24A single mutant still displayed binding ability, although with an impairment in the iron transport kinetics [23]. One possible explanation of these results is that other residues might play an indirect role in iron binding by contributing to the negative electrostatic potential of the cavity (Figure 5). The overall negative electrostatic potential generated by the acidic residues inside the cavity of BdFpn appears to be the key driving force for the observed spontaneous metal influx, whereas stabilization of the iron in the binding site requires direct involvement of specific coordinating residues (Figure 7 and Appendix A).

In the preprint posted on bioRxiv, Billesbølle and coworkers [22] demonstrated that, in hFpn, the Fe^2+^ ion interacts with side chains of H507, D325, and D504 and the backbone carbonyl of T320 [22]. The human ferroportin residue D325 corresponds to H261 (TM7) in BdFpn, and indeed previous work has shown that H261 is critical for metal binding in BdFpn [23,24]. In this context, the N-terminal domain could provide the early binding site composed of D24, D134, S130, E203 residues. Arrival of the iron at this site may induce conformational changes, including the local shifts in TM7 (H261) and TM10 along with the relative rotation of the N-terminal and C-terminal domains. The transition to the occluded state would thus facilitate the formation of a novel binding site in the unwound segment of TM7 where H261 is located.

## 4. Materials and Methods 

The BdFpn structures (apo-inward facing, PDB ID: 5AYO; apo-outward facing, PDB ID: 5AYN) [24] were firstly subjected to a modelling procedure through the UCSF Chimera program [29], to introduce a loop between T222 and S240, unresolved in the crystal structures. This task was accomplished taking advantage of the “Model loop” plugin of Chimera, a graphical interface to the Modeller engine [30] that models the loops using existing segments, refined by generating additional possible conformations. The resulting models were then subjected to 100 steps of energy minimization to remove any unfavorable interaction within the modelled loop. The minimized structures were then embedded in a lipid bilayer, mimicking the membrane composition of *Bdellovibrio bacteriovorus* through the membrane builder function of the CHARMM-GUI web server [31], a graphical user interface to generate molecular simulation systems. The membrane bilayer composition was the following: dilauroylphosphatidylcoline (DLPC), lower leaflet 221 molecules, upper leaflet 238 molecules; dilauroylphosphatidylethanolamine (DLPE) lower leaflet 42 molecules upper leaflet 39 molecules. The CHARMM-GUI output, consisting in the protein–membrane complex surrounded by water molecules, was used as input for the tLeap module of the AmberTools 15 suite [32], to generate the topology and coordinates files required to perform the MD simulations. Unless otherwise stated, all simulations were conducted using the following protocol. The structures were parameterized using the AMBER ff14sb force field [33] for the protein and the lipid14 force field [34] for the lipids. The protein–membrane complex was solvated using TIP3P water molecules, neutralizing the system charges with the correct number of monovalent counterions placed in electrostatically favored positions. Then, 200 Fe^2+^ ions, when present, were randomly placed in the solvent (and neutralized with monovalent counterions), using the divalent ion parameters from the Compromise set (CM set), which reproduced the experimental relative hydration free energies and coordination number values, for Particle Mesh Ewald and TIP3P water model from Li et al. [35]. The system was energy minimized for 5000 steps, applying position restraints on the protein and lipid bilayer to remove any clashes and unfavorable interactions occurring within the components of the system. After energy minimization, several MD runs were performed to equilibrate the systems, in which the temperature was gradually increased from 0 to 303 K using the NVT ensemble, keeping the lipid and protein restrained with a force constant of 2.5 kcal·mol^−1^·Å^−2^. In the subsequent steps, the harmonic restraints were gradually reduced, for the proper positioning of lipid molecules around the protein. Then, 100 ps equilibrium simulations were performed without restrains followed by a 250 ps run using the NPT ensemble before starting the production phase. Simulations were then carried out using periodic boundary conditions, with a cut-off radius of 10.0 Å for the non-bonded interactions, which were smoothed through a cut-off function beyond 8.0 Å. The neighbors pair-lists were updated every 10 steps with an inclusion distance of 13.5 Å. The electrostatic interactions were calculated every two steps (4.0 fs) using the Particle Mesh Ewald method with 1.0 Å grid spacing [36]. The simulations were carried out at a constant temperature of 303 K, using Langevin dynamics with a damping coefficient of 5 ps as coupling coefficient to be applied to all atoms [37]. For the pressure control the Langevin piston Nose–Hoover method, a combination of the Nose–Hoover constant pressure method [38] with piston fluctuation control implemented using Langevin dynamics [39], was used specifying the desired pressure at 1.01325 bar (1 atm), the oscillation period at 100 fs, the decay times of the piston with the barostat damping time scale at 50 fs and the temperature of the piston at 303 K. The SHAKE [40] algorithm was applied to constrain the protein hydrogen atoms, while the SETTLE algorithm [41] was applied to constrain the water atoms. Atomic coordinates were saved every 1000 steps (2 ps). Each model was simulated for 200 ns and the atomic positions were saved every 1000 steps (i.e., 2.0 ps) for the subsequent analyses. To test reproducibility, for each system, three independent MD simulations were performed assigning random velocities to the atoms before the production phase. RMSD, RMSF, PCA, cluster and distance analyses were carried out through the GROMACS 4.6.7 analysis tools [42]. CPPTRAJ was used to measure the average density of the bilayer components and the thickness of the membrane along the trajectory. The results are shown in Appendix A and indicate that the membrane system is stable in all the systems simulated [43]. VMD plugins were employed to analyze salt bridges (cut-off distance between the oxygen and nitrogen atoms of the respective residues: 5.0 Angstroms) and hydrogen bonds. A hydrogen bond has been considered as formed between an atom with a hydrogen bonded to it (the donor, D) and another atom (the acceptor, A) provided that the distance D-A was less than the cut-off distance (default 3.0 Angstroms) and the angle D-H-A was less than the cut-off angle (default 20 degrees) [44].

## 5. Conclusions

MD simulations of BdFpn allowed us to characterize the conformational dynamics of the protein and to identify key residues involved in the initial encounter with iron and conformational switch of the transporter. From this viewpoint, it must be considered that classical MD simulations are unable to simulate the actual coordination event of a metal ion by a protein, given that the spin state of the metal and coordination geometry effects cannot be included in the ion parametrization normally employed in these kind of simulations. The analyses revealed that the inward-facing state is less stable than the outward-facing conformation, in line with the results obtained for other MFS transporters [45]. Salt-bridge reorganization appears to constitute an important set of molecular events facilitating the conformational change in the transporter. The optimization of these interaction networks seems to have a role in the reduced conformational dynamics of the transporter in order to achieve a productive transport cycle. Indeed, salt-bridges formation and breakage seem to be a common feature of the transport mechanism of a variety of MFS transporters [2]. The importance of these interactions is also highlighted by the involvement of hFpn orthologous residues in hereditary iron disorders [27]. The results presented here well correlate with available data on MFS transporters in general, and on Fpn transporters in particular. They appear to be successful not only at describing the global movements of BdFpn but also in defining key residues likely involved in the binding and translocation of iron through the transporter. These studies are useful for our understanding of BdFpn transport mechanism and should be useful also in future biochemical, mutagenesis and functional studies of hFpn. As a final comment, steered MD coupled with Potential of Mean Force calculations of an iron atom, in the different conformational states of the protein could in principle give more valuable information with respect to the classical MD simulations described in this work. Unfortunately, the picture of the conformational transitions taking place during iron transport is still incomplete. In fact, there is no structure available for the occluded state of the protein with iron bound. The precise identity of the iron ligands is therefore unknown. Thus, steered MD simulations will only be possible in the future if the occluded state structure is solved.

## Figures and Tables

**Figure 1 ijms-21-06785-f001:**
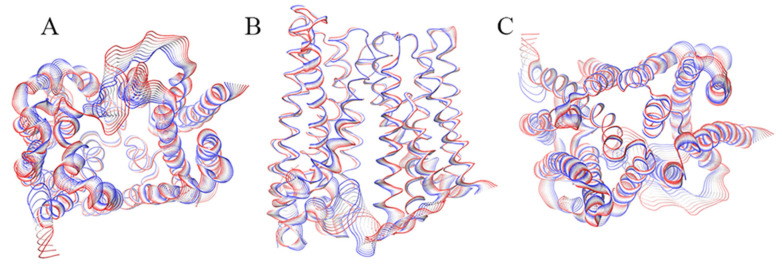
Graphical representation of the motion projections identified along the first eigenvector on the three-dimensional structure of the BdFpn. (**A**) Inward side view; (**B**) View along the membrane plane; (**C**) Outward side view. The width of the ribbon, generated by the flanking tubes, indicates the amplitude of the motion while the transition from red to blue indicates the motion direction.

**Figure 2 ijms-21-06785-f002:**
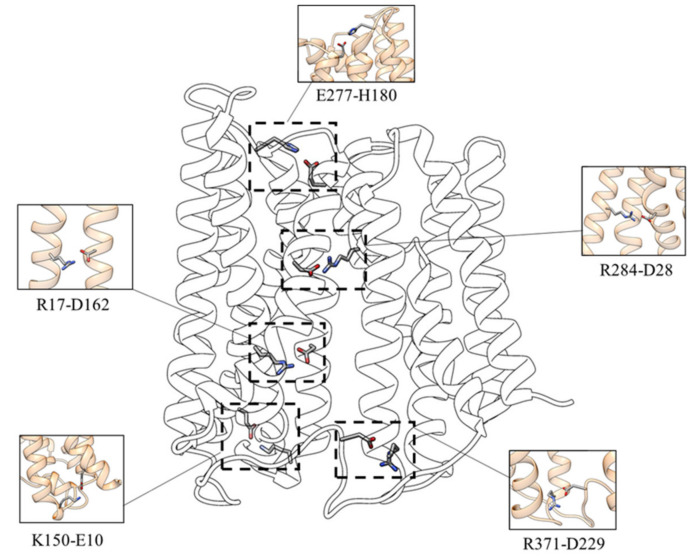
Salt bridges network involved in the stability of the inward-facing state.

**Figure 3 ijms-21-06785-f003:**
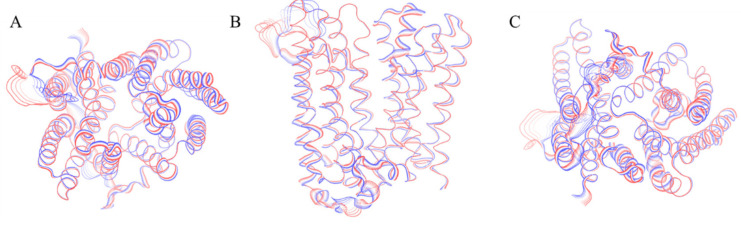
Graphical representation of the motion projections identified along the first eigenvector on the three-dimensional structure of the BdFpn. (**A**) Top view; (**B**) Front view; (**C**) Bottom view. The width of the ribbon, generated by the flanking tubes, indicates the amplitude of the motion while the transition from red to blue indicates the motion direction.

**Figure 4 ijms-21-06785-f004:**
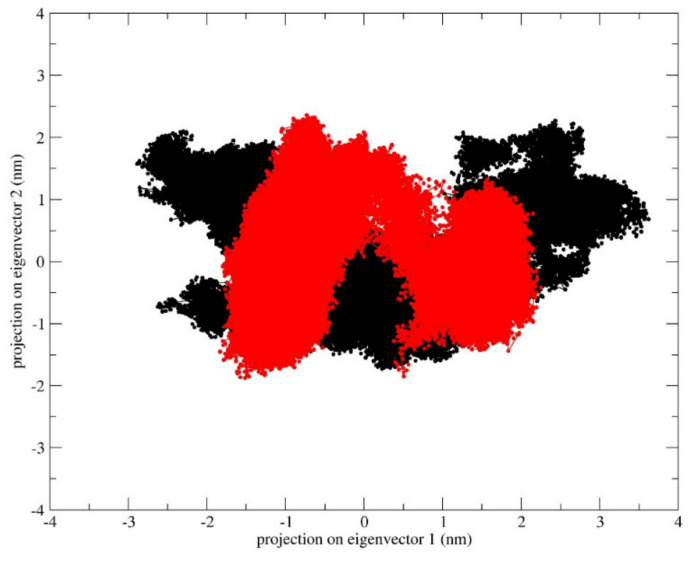
Projections of the inward-facing state (black) and outward-facing state (red) trajectories on the common essential subspace, as defined by the two principal eigenvectors.

**Figure 5 ijms-21-06785-f005:**
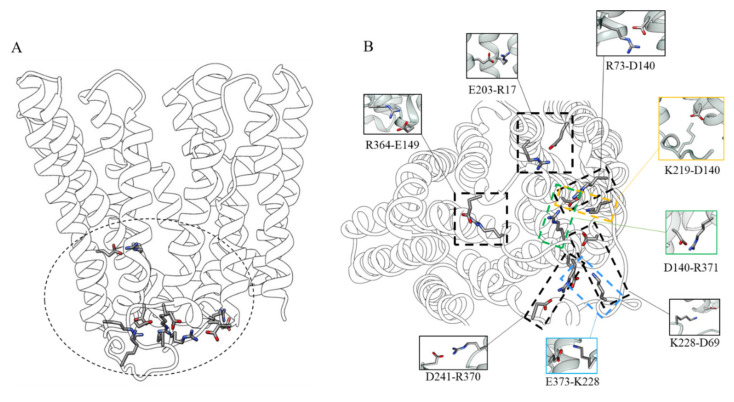
(**A**) Residues interaction network involved in the stability of the outward-facing state viewed along the membrane plane; (**B**) View of the intracellular-side with focus on the residue interactions.

**Figure 6 ijms-21-06785-f006:**
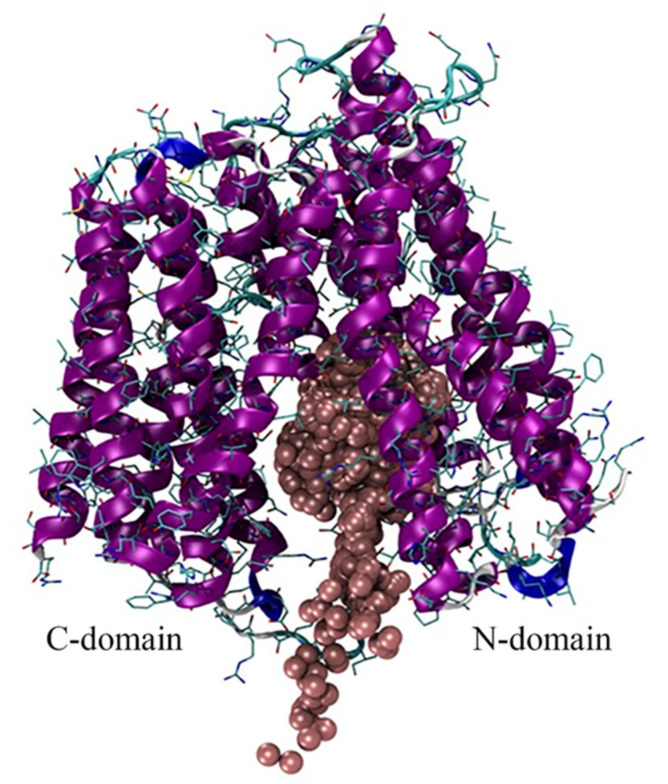
Superimposition of representative snapshots of the Inward_Fe trajectory. Iron ions are shown as brown spheres.

**Figure 7 ijms-21-06785-f007:**
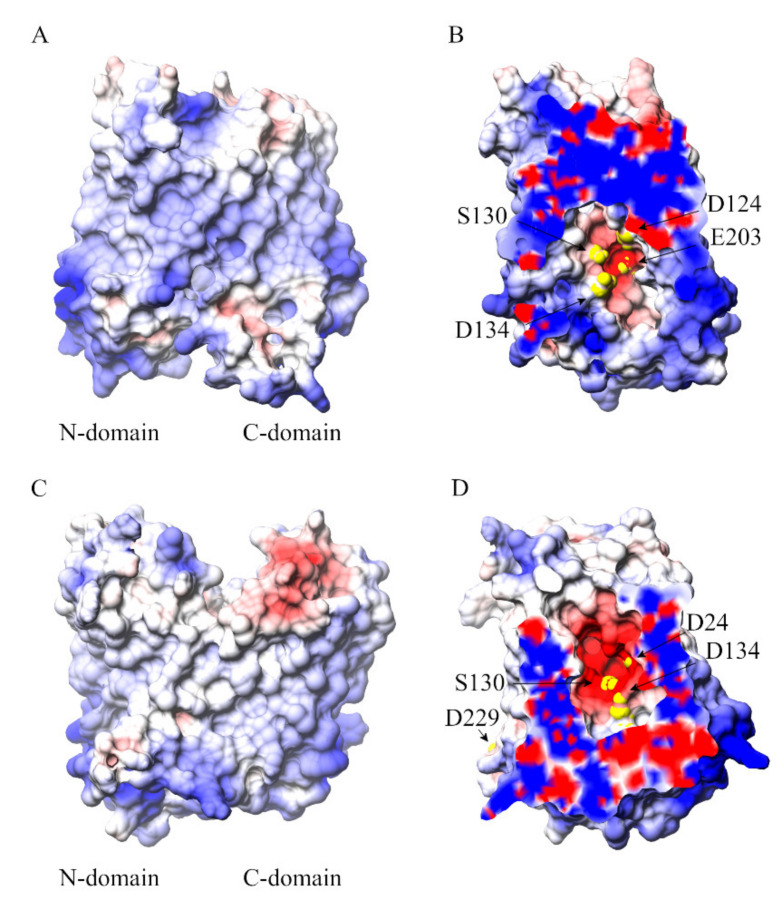
(**A**) View along the membrane plane of the overall structure of the inward-facing BdFpn structure, and (**B**) close-up of the negatively charged pocket turned by 90 degrees with respect to panel A, with molecular surface colored by electrostatic potential values from red (−5 kT/e) to blue (+5 kT/e); (**C**) View along the membrane plane of the overall structure of the outward-facing BdFpn structure, and (**D**) close-up of the negatively charged pocket turned by 90 degrees with respect to panel C, with molecular surface colored by electrostatic potential values from red (−5 kT/e) to blue (+5 kT/e). Residues identified in the iron-occupancy analysis of the Inward_Fe trajectories are depicted as yellow spheres. The electrostatic potential has been calculated using APBS web server (http://server.poissonboltzmann.org/) [26].

**Table 1 ijms-21-06785-t001:** Systems subjected to MD simulations.

System	Duration (ns)	Conformational State	Iron	Number of Simulations
Inward_Apo	200	Inward-facing	-	3
Outward_Apo	200	Outward-facing	-	3
Inward_Fe	200	Inward-facing	+	3

**Table 2 ijms-21-06785-t002:** Summary of the iron occupancy around residues during the simulations.

Residue	Occupancy
D24	18.2%
D134	33.3%
S130	18.3%
E203	11.1%
D229	13.8%

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
