# Peer review of "Dynamical Behavior of the Human Ferroportin Homologue from Bdellovibrio bacteriovorus: Insight into the Ligand Recognition Mechanism"

_ijms, 2020, doi:10.3390/ijms21186785_

Round 1

Reviewer 1 Report

The submitted work describes molecular dynamic investigation of the human ferroportin homologue. Experimental data on structure, and particularly the mechanism of iron transport of this transmembrane protein, are scarce. In this respect, the present work represents an important contribution to the understanding of how ferroportin regulate iron transport process.

The work has been done with care, and I can recommend it for publication in International journal of Molecular Sciences after just minor revision.

The authors claim (lines 306-310) they have identified critically important residues for iron binding in the ferroportin "pocket". I'm wondering how reliable are results obtain through MD simulation in this case. Iron complexes are known for having different spin states, where the geometry (and energy) of complexes is dependent on spin state. Are the parameters for iron in MD simulations reliable to address this issue, when even DFT sometime fails. I would like to see some comment on that. 

Author Response

Point-to-point reply

Reviewer 1

The submitted work describes molecular dynamic investigation of the human ferroportin homologue. Experimental data on structure, and particularly the mechanism of iron transport of this transmembrane protein, are scarce. In this respect, the present work represents an important contribution to the understanding of how ferroportin regulate iron transport process.

The work has been done with care, and I can recommend it for publication in International journal of Molecular Sciences after just minor revision.

The authors claim (lines 306-310) they have identified critically important residues for iron binding in the ferroportin "pocket". I'm wondering how reliable are results obtain through MD simulation in this case. Iron complexes are known for having different spin states, where the geometry (and energy) of complexes is dependent on spin state. Are the parameters for iron in MD simulations reliable to address this issue, when even DFT sometime fails. I would like to see some comment on that.

We thank the Reviewer for this comment that allows us to better explain in the manuscript the aim of our MD simulations. We were interested in simulating the initial “encounter” of the iron ions with the protein binding site before the ion itself gets coordinated by protein residues. Indeed, as correctly pointed out by the Reviewer, classical MD simulations are unable to simulate the actual coordination event of a metal ion by a protein given that the spin state of the metal and coordination geometry effects cannot be included in the simple “charged sphere” representation of the metal usually employed in this kind of simulations. However, the iron ions were parametrized using the Compromise set (CM set) for Particle Mesh Ewald and TIP3P water model (from Li et al., 2013), which was proven to reproduce the experimental relative hydration free energy and coordination number values for divalent ions. A comment along these lines has been inserted in the revised version of the manuscript (beginning of the Conclusions Section) and details have been added in the Materials and Methods Section (pag. 11, bottom part).

Reviewer 2 Report

In this work, Tortosa et al have performed several MD simulations of the human ferroportin homolog from the Gram-negative bacterium Bdellovibrio bacteriovorus in two major conformational states, and also in the presence of high Fe2+ concentration to try to identify key structural determinants that could regulate iron-binding and translocation through this protein’s channel. In overall, the work is well described, but in order to further suggest this paper for publication in IJMS, I would appreciate that the authors could give further details regarding the following points:

  1. In neither the simulations presented in this paper, the authors mentioned quality measurements regarding the membrane where the protein is embedded. This must be done, and at least the results should be placed in SM and a small phrase should mention these tests in the paper.
  2. The authors should also explain why they chose as membrane model DLPC lipids.
  3. An important issue regarding the simulations of the protein in the presence of Iron is the simulation parameterization of these ions. Therefore, it would be important for readers to have more detailed information regarding these ions.
  4. In the simulations in the presence of iron atoms, I could not really understand how they could perform the simulations with such system charge imbalance present in PME simulations. Further details must be given regarding these simulations, and also it must be proven that the membrane where this system is embedded is also stable.
  5. In order to further characterize the simulations in the presence of iron, the authors should also mention why they did not perform these simulations for the Outward conformation. Actually, I believe that the work could be much more complete if such simulations could also be performed.
  6. Figure 7 images could be redesigned since this mesh-like representation is not the most adequate to show the electrostatic potential mapping. A slick and uniform surface could be used in such images, making, therefore, clearer for the reader to analyze them.
  7. In order to complement figure 7, the electrostatic potential images should be performed for both conformational states simulated in the paper. This is only done for one, and the additional information could also be important to understand the structural charge features, regulating the permeation of the Fe atoms through the protein.
  8. As a suggestion, I believe that performing Steered MD coupled with Potential of Mean force calculations of an iron atom, in the different conformational states of the protein studied in this work could give must more valuable information than the simulation presented in this work. The energetical information acquired from these simulations could complement the simple MD simulations presented by the authors.

Therefore, based on the previously mentioned points, I would appreciate that the authors could comment on the different aspects I have raised. Only, after this is done, I will be able to consider this work to be accepted for publication.

Author Response

Point-to-point reply

Reviewer 2

In this work, Tortosa et al have performed several MD simulations of the human ferroportin homolog from the Gram-negative bacterium Bdellovibrio bacteriovorus in two major conformational states, and also in the presence of high Fe2+ concentration to try to identify key structural determinants that could regulate iron-binding and translocation through this protein’s channel. In overall, the work is well described, but in order to further suggest this paper for publication in IJMS, I would appreciate that the authors could give further details regarding the following points:

  1. In neither the simulations presented in this paper, the authors mentioned quality measurements regarding the membrane where the protein is embedded. This must be done, and at least the results should be placed in SM and a small phrase should mention these tests in the paper.

As suggested by the Reviewer, we have added the quality measurements data regarding the membrane (i.e., bilayer thickness along the trajectory and average membrane density) in the Supplementary Materials section (Figures S6-11). Also, a mention to these tests has been added in the Materials and Methods Section of the revised manuscript.

  1. The authors should also explain why they chose as membrane model DLPC lipids.

As specified in the Methods section of the manuscript, the membrane model was made up by a mixture of dilauroylphosphatidylcoline (DLPC) and dilauroylphosphatidylethanolamine (DLPE) lipids. This was the best option we had to mimic the Bdellovibrio bacteriovorus membrane, which is made up by a mixture of phosphatidylethanolamine- and phosphatidylglycerol-containing phospholipids, given that validated parameters in AMBER Lipid14 force field are available for PC and PE.

  1. An important issue regarding the simulations of the protein in the presence of Iron is the simulation parameterization of these ions. Therefore, it would be important for readers to have more detailed information regarding these ions.

We agree with the Reviewer, and therefore details on the iron parametrization, using the Compromise set (CM set) for Particle Mesh Ewald and TIP3P water model (from Li et al., 2013), which was proven to reproduce the experimental relative hydration free energy and coordination number values for divalent ions, have been added to the Materials and Methods section of the revised manuscript (pag. 11, bottom part).

  1. In the simulations in the presence of iron atoms, I could not really understand how they could perform the simulations with such system charge imbalance present in PME simulations. Further details must be given regarding these simulations, and also it must be proven that the membrane where this system is embedded is also stable.

We understand the Reviewer’s concern regarding the charge of the system in presence of iron atoms, however the system was neutralized with monovalent counterions. This has been better clarified in the Materials and Methods section of the revised manuscript. Further, the results of quality measurements regarding the membrane (i.e., bilayer thickness along the trajectory and average membrane density) have been added in Supplementary Materials section (Figures S6-11).

  1. In order to further characterize the simulations in the presence of iron, the authors should also mention why they did not perform these simulations for the Outward conformation. Actually, I believe that the work could be much more complete if such simulations could also be performed.

Given the fact that ferroportin is responsible for iron efflux from cells, we were interested in simulating the initial “encounter” of the iron ion with the protein’s binding site before the ion itself gets coordinated by protein residues. This is why we simulated only the inward-facing conformation of the protein in the presence of iron. A brief reference to this choice has been added in the initial part of the Results Section of the revised manuscript. However, we will take into account the Reviewer’s advice for future development of our work.

  1. Figure 7 images could be redesigned since this mesh-like representation is not the most adequate to show the electrostatic potential mapping. A slick and uniform surface could be used in such images, making, therefore, clearer for the reader to analyze them.

We thank the Reviewer for the advice. The figure 7 has been modified as suggested.

  1. In order to complement figure 7, the electrostatic potential images should be performed for both conformational states simulated in the paper. This is only done for one, and the additional information could also be important to understand the structural charge features, regulating the permeation of the Fe atoms through the protein.

As stated above, the figure 7 has been modified according to the Reviewer’s suggestions.

  1. As a suggestion, I believe that performing Steered MD coupled with Potential of Mean force calculations of an iron atom, in the different conformational states of the protein studied in this work could give must more valuable information than the simulation presented in this work. The energetical information acquired from these simulations could complement the simple MD simulations presented by the authors.

The simulations suggested by the Reviewer would certainly be very interesting and informative if the structure of all the relevant conformational states of the protein were available. Unfortunately, we miss an important “piece of the puzzle” because there is no structure available for the occluded state of the protein with iron bound. The precise identity of the iron ligands is therefore unknown and thus we are afraid that Steered MD simulations cannot be carried out in this case.

Therefore, based on the previously mentioned points, I would appreciate that the authors could comment on the different aspects I have raised. Only, after this is done, I will be able to consider this work to be accepted for publication.

Round 2

Reviewer 2 Report

Taking into account the reply the authors gave to my questions and comments I am now satisfied with the presented version of the work. My only comment to the authors is only regarded to the last point I raised (point 8). I believe that the reply they gave to my question should be included in the final discussion, pushing the evaluation of this type of simulations to a future work.

Author Response

As suggested by the Reviewer, the following sentences have been added to the last part of the Conclusions section: "As a final comment, steered MD coupled with Potential of Mean force calculations of an iron atom, in the different conformational states of the protein could in principle give more valuable information with respect to the classical MD simulations described in this work. Unfortunately, the picture of the conformational transitions taking place during iron transport is still incomplete. In fact there is no structure available for the occluded state of the protein with iron bound. The precise identity of the iron ligands is therefore unknown. Thus, steered MD simulations will only be possible in the future if the occluded state structure will be solved."